# A note on the statistical evidence for an influence of geomagnetic activity on JRA-55 northern hemisphere seasonal-mean stratospheric temperatures

Nazario Tartaglione[1,2], Thomas Toniazzo[1,2], Yvan Orsolini[2,3,4], and Odd Helge Otterå[1,2]

[1]NORCE Climate, Bergen, Norway
[2]Bjerkness Centre for Climate Research, Bergen, Norway
[3]Birkeland Centre for Space Science, University of Bergen, Bergen, Norway
[4]Norwegian Institute for Air Research, Kjeller, Norway

**Correspondence:** Nazario Tartaglione (nazario.tartaglione@norceresearch.no)

**Abstract.** We employ JRA-55, a recent second-generation global reanalysis providing data of high-quality in the stratosphere, to examine whether a distinguishable effect of geomagnetic activity on northern hemisphere stratospheric temperatures can be detected. We focus on how the statistical significance of stratospheric temperature differences may be robustly assessed during years with high and low geomagnetic activity. Two problems must be overcome. The first is the temporal autocorrelation of the data, which is addressed with a correction of the t-statistics by means of the estimate of the number of independent values in the series of correlated values. The second is the problem of multiplicity due to strong spatial autocorrelations, which is addressed by means of a false discovery rate (FDR) procedure. We find that the statistical tests fail to formally reject the null hypothesis, i.e. no significant response to geomagnetic activity can be found in the seasonal-mean northern-hemisphere stratospheric temperature record.

## 1   Introduction

There is a large interest in the potential climate impact of geomagnetic activity. One of the main mechanisms by which geomagnetic activity is thought to affect the middle atmosphere is through the production of nitrogen oxides ($NO_x$'s), either by the continuous precipitation of auroral electrons penetrating into the lower thermosphere (Sinnhuber et al., 2012) or by the more episodic precipitation of higher energy electrons into the mesosphere (Andersson et al., 2014; Päivärinta et al., 2016). Downward transport from the mesosphere to the stratosphere in winter results in increased availability of $NO_x$ in the dark polar stratosphere, where it is long-lived. $NO_x$ can catalytically reduce ozone concentrations as the sun returns (Brasseur and Solomon, 1986; Callis et al., 2005), and thus alter radiative heating rates, with potential observable impacts on stratospheric temperatures and possible implications also for surface air temperature (SAT). The amount of $NO_x$ in the middle atmosphere during late winter and spring depends on the cumulative effect of geomagnetic activity over the preceding months on the $NO_x$ reservoir (Jacob, 1999). Stratospheric $NO_x$ concentrations however also depend on the magnitude of the downward transport from this reservoir, and are thereby affected by internal variability of the atmospheric circulation from year to year, especially in the Northern Hemisphere (NH) (Funke et al., 2005; Randall et al., 2006; Päivärinta et al., 2016).

The impact of energetic electron precipitation (EEP) driven by geomagnetic activity on $NO_x$ and ozone concentrations has been well-documented after detailed satellite studies were carried out in the early 2000's (Funke et al., 2005; Randall et al.,
2005). Several recent studies (Baumgaertner et al., 2010; Bucha, 2014; Lu et al., 2008; Seppälä et al., 2009, 2013) suggest a significant signal associated with geomagnetic activity in the observed climate. However, there remains considerable uncertainty regarding the precise attribution of such signal, and the existence of a direct link between EEP and stratospheric and tropospheric temperatures has remained controversial. Among others, the study of Seppälä et al. (2009), henceforth S09, in particular, claims to find a significant, direct relationship between SAT and geomagnetic activity based on reanalysis data from
the European Centre for Medium Range Weather Forecasts (ECMWF). In essence, S09 finds that the hypothesis that geomagnetic activity influences the SAT is supported by reanalysis data, whereas the null hypothesis that the SAT is not influenced by the geomagnetic activity at all is rejected. S09 compares seasonal SAT in years with high and low geomagnetic activity, and also considered the separate effect of the variation in solar irradiance associated with the 11-year heliomagnetic cycle.

The selection of years in S09 was based on two indices, Ap and f10.7. Ap (Rostoker, 1972) provides a measure for daily aver-
35 age level of geomagnetic activity. To account for cumulative effect of $NO_x$ production, transport and diffusion processes, Ap was commonly averaged over 4 months from late autumn to winter (Seppälä et al., 2009; Funke et al., 2014; Tomikawa, 2017). In particular, S09 used Ap averaged between October and January to define winters of high and low geomagnetic activity in the northern hemisphere. The second index, f10.7 (https://www.swpc.noaa.gov/phenomena/f107-cm-radio-emissions), is an indicator of the phase and intensity of the solar cycle. By compositing separately on the basis of Ap and f10.7, S09 obtained
different samples of seasonal-mean data for years with high geomagnetic activity and for years with low geomagnetic activity. They then computed the SAT differences of the seasonal means (DJF, MAM, JJA, SON) between the two samples, and employed a t-test based on the set of daily-means (Seppälä, personal communication) used to compute the seasonal averages to discriminate against a null hypothesis of no effect. As a consequence of such procedure, S09's claimed of significance is marred by the presence of very strong temporal and spatial autocorrelation within the samples.

In this paper, we revisit the S09 hypothesis by adopting a rigorous methodology for significance testing on strongly autocorrelated data. We focus on wintertime stratospheric temperatures between 200 hPa and 1 hPa, a pre-requisite for possible surface impacts associated with EEP-related changes in ozone concentrations. Although our analysis is focused, in this paper on stratospheric temperature, we look at all the levels present in the dataset. We show that statistical testing appropriate to the data at hand is a crucial step in any analysis purporting to demonstrate an observed climate signal of geomagnetic activity.

Data and methods are described in section 2, including a discussion on the problem of autocorrelation in time and space. In section 3, the results obtained by applying the t-test to the stratospheric temperatures are shown. The analysis is applied to four different cases: with no correction at all, with the temporal and the spatial autocorrelation correction applied separately, and with both the corrections applied. In Section 4 conclusions are drawn.

## 2 Data and Methods

### 2.1 Data

To analyze the possible impact of geomagnetic activity in the stratosphere, we use the Japanese 55-year reanalysis (JRA-55) covering more than 55 years, extending from 1958 to the present (Kobayashi et al., 2015). Due to the selection of cases of high and low geomagnetic activity as in S09, only data up to 2006 is used here. In JRA-55 reanalysis ozone is used interactively in the radiation code, although it is treated differently in the pre- and post-1979 satellite era. This is an important asset for JRA-55 since the EEP will primarily affect $NO_x$ and ozone, and this feature is not commonly found in other reanalysis systems, such as the ECMWF reanalyses. Older generation reanalyses tend to suffer from temporal inhomogeneities because of the sequential introduction of new satellite data during the assimilation period, especially in the SH as shown recently by Long et al. (2017). For these various reasons, we restricted our analysis to the recent JRA-55 reanalysis. Tomikawa (2017) also used the JRA-55 reanalyses to investigate the signature of geomagnetic activity, but focused exclusively on the SH. He found a temperature signal in the upper stratosphere, but only in July. The S09 selection shown in Table 1 is used to compute the significance of the seasonal differences. The criteria used to select the different years are based on the Ap and f10.7 values, and are the same as used by S09. The definition of high and low geomagnetic activity is the same as S09. We hence investigate the potential signatures on stratospheric temperature during the same winters and in the following seasons of the same calendar year as S09 did for SAT. The set of data is denominated N1 as in S09 (Table 1).

### 2.2 Data autocorrelation and statistical significance

S09 computed the SAT differences of the seasonal means (DJF, MAM, JJA, SON) between those selected high Ap and low Ap years, and employed the Welch's t-test (hereafter only t-test) to assess the likelihood of the differences given a null hypothesis of no effect. Such a test assumes a statistical model in which observations are normally distributed and statistically independent. In particular, the t-test is sensitive to the temporal autocorrelation or serial correlation within the samples. When serial correlation is not taken into account in the data, statistically significant differences in two means, which may not be different at all, are found more frequently than expected (Zwiers and von Storch, 1995). S09's analysis is affected by this problem, because the authors used daily-mean data in their t-test, which are highly autocorrelated in time. As seasonal averages can still suffer from temporal autocorrelation, the serial dependence is checked by means of the Durbin-Watson test (Durbin and Watson, 1950). While the serial correlation, in general, is reduced from seasonal averaging, it can still persist, especially in summer. To deal with such serial correlation a correction is applied as suggested by Zwiers and von Storch (1995). The temporal autocorrelation is not the only potential caveat that needs to be considered when testing a hypothesis. When performing a significance test simultaneously on many samples one will at some point find statistically significant temperature differences simply by accident. Unfortunately, the dominant approach to the multiplicity problem is generally to test the single grid points and then to report them as "significant" when the null hypothesis is locally rejected (Wilks, 2016). Sometimes temporal and spatial autocorrelation is not addressed at all, but , there are some exceptions. Maliniemi et al. (2014), for instance, trying to find a relationship between solar activity and surface air temperature dealt with temporal and spatial autocorrelation using a

Monte Carlo approach. To overcome this multiplicity problem in our analysis, we apply the false discovery rate controlling introduced by Benjamini and Hochberg (1995) and proposed in the atmospheric sciences by (Wilks, 2006, 2016).

## 2.3 Accounting for temporal autocorrelation

The t-test is a widely used method for hypothesis testing within the climate community. It is however well known that the t-test, which assumes a statistical model where observations are statistically independent and it is widely, but incorrectly, believed that the t-test is valid only for normally distributed outcomes. Several authors (Efron, 1969; De Winter, 2013; Poncet et al., 2016) have shown that the t-test is suitable under symmetric, not necessarily normal, and asymmetric distributions. The t-test is sensitive to time autocorrelation or serial correlation within the samples. The effect of serial correlation is, usually, to make

comparisons of means too liberal. The null hypothesis assuming equal means is hence rejected more frequently than expected. Two separate reasons favor the use of seasonal-mean data instead of daily-mean data. The first reason is that any influence of EEP on temperature is expected to accumulate over seasonal time scales. The second reason is that daily temperatures are strongly serially correlated, whereas seasonal data have less correlation between two consecutive years for instance. In fact, one of the causes of the serial correlation is that the variable of interest varies seasonally. Nevertheless, even for seasonal means it

is important to account for serial correlations, as there may be other causes leading temporal autocorrelation, including persistence. Fig. 1a that shows the results of the Durbin-Watson test (Durbin and Watson, 1950) applied at the seasonal temperatures at 5 hPa. Similar pictures can be obtained by plotting the lag-1 autocorrelation (Fig. 1b), but the Durbin-Watson test, which is a classical test to check whether data are serially correlated, is better, compared to including the lagged response, as it tests for autocorrelation in the residuals and it is suitable when in time series there are trends or seasonal patterns. When data are

serially correlated, the test gives values close to zero, whereas when data are not correlated the test statistic values, as a rule of thumbs, are in the range of 1.5 to 2.5. There is also the possibility of serial anti-correlation: in such a case, the value would be above 2.5, but this situation was not found in our study.

During the winter and spring seasons, the data generally do not have a very strong temporal autocorrelation, and the t-test can be applied with a lower risk of obtaining false positive outcomes. There are some regions where the temporal autocorrelation

still persists, such as over North America. Local local higher autocorrelation values during other seasons can also be a result due also to low frequency variance caused by large scale teleconnections (Madden, 1977). During the summer season data are very autocorrelated - and to a large extent also in autumn, but they will be analyzed in any case as it is worthwhile as well to show how the procedure used to assess the possible impact of the geomagnetic activity responds to serially correlated data. In general, autocorrelation is mainly due to persistence of temperature patterns year by year. For instance, this is the case for

example of the large value of temperature autocorrelation found during the summer season. However, we cannot exclude other causes, including a possible impact of the solar activity.

Serial correlation can be corrected for by adopting, for example, the strategy suggested by Zwiers and von Storch (1995). This procedure is valid under the assumption that the time series, from which the data are sampled can be modelled as an autoregressive process of order 1 or AR(1). Vyushin et al. (2012) have shown that the AR(1) representation fits modeled stratospheric

temperature data very well according to standard goodness of fit tests. Seidel and Lanzante (2004) found a similar result with

temperature observed by radiosondes and satellites.

If EEP has a cumulative impact during the different seasons, it has to be shown that the means of two subsets with high (H) and low (L) Ap values from the set N1 must be different.

To test the null hypothesis $H_0 : \mu_H = \mu_L$ with the t-statistics at the 5% significance level one let's apply the t-test under the condition that the standard deviation is scaled by the equivalent sample sizes $m_e$ and $n_e$ that can be computed, by:

$$n_e = n \left( \frac{1 - \rho_1}{1 + \rho_1} \right) \tag{1}$$

where $n$ is the original size of one out of two samples and $\rho_1$ is the parameter of the AR(1) process representing the autocorrelation at lag 1; and similar for $m_e$. The t-test is then corrected in the following way:

$$t = \frac{\bar{H} - \bar{L}}{s \left( \frac{1}{\sqrt{m_e}} + \frac{1}{\sqrt{n_e}} \right)} \tag{2}$$

where $\bar{H}$ and $\bar{L}$ are the sample averages and $s^2$ is the pooled variance

$$s^2 = \frac{\sum_{i=1}^{m} \left( H_i - \bar{H} \right)^2 + \sum_{i=1}^{m} \left( L_i - \bar{L} \right)^2}{m + n - 2} \tag{3}$$

## 2.4 Accounting for spatial autocorrelation

Spatial autocorrelation produces the so-called multiplicity problem, which arises when testing a statistical hypothesis on many
samples (the domain's grid points, in our case) simultaneously. A single hypothesis test allows for a null hypothesis and an alternative hypothesis. The alternative hypothesis will be favored when an extreme value, usually with a probability (called value) that is less than 5% is found (Wilks, 2016). Making a statistical test on multiple points, for example within a spatial domain, means that more realizations will be available and there will be many grid points where one is more likely to reject the null hypothesis. In an ideal situation, where the value is set to 0.05 and each point is statistically independent of the others, it
is expected to find that 5% of the points will be statistically significant by accident. The situation is worse when the grid points are correlated, as is often the case when analyzing meteorological and climate data. This problem, known in the literature as the multiplicity problem, has been encountered in several studies, although most of the studies in the atmospheric science have not properly addressed the issue yet (Wilks, 2016). Some solutions have been proposed, each having their own advantages and disadvantages. Wilks (2016) gives a brief historical outline and shows different solutions to this problem. One technique to
address this issue is by using the false discovery rate (Benjamini and Hochberg, 1995). According to Wilks (2006, 2016) the false discovery rate is the expectation of the fraction of true null hypothesis rejections among all the rejections and it is the best available approach to analyze multiple hypothesis test results, even when those results are mutually correlated.

As stated by Wilks (2016) the FDR procedure requires smaller values to reject the local null hypothesis arising the standard of the test. For the sake of the reader we will describe the FDR algorithm as described in Wilks (2016). The algorithm operates on the collection of $H_0 : \mu_H = \mu_L$ values from $m_e$ (number of grid points) local hypothesis tests $p_i$, with $i = 1, ..., N$, which are sorted in ascending order. Rejection of the test happens when the $p_i$ values are not larger than a threshold level $p_{FDR}$ that is a function of the distribution of the sorted $p_i$ values. More specifically to define which values pass the test the following formula is used:

$$\left[ p_i : p_i \leqslant \alpha_{FDR} \left( \tfrac{i}{N} \right) \right]$$

where $\alpha_{FDR}$ is the chosen FDR control level that here is taken equal to 0.05. For a given value of $\alpha_{FDR}$, the largest value of $i$, such that $p_i \leqslant \alpha_{FDR} \left( \tfrac{i}{N} \right)$ defines the threshold below which the local null hypotheses are rejected. the largest value of i, such that

## 3   Results

### 3.1   Stratospheric levels

We start with the application of the t-test on 5 hPa temperature (Fig. 2), which represents the level where the statistically significant area is the largest among all the examined pressure levels. There are large areas with statistically significant temperature difference at 5% level especially during winter and summer.

At 5 hPa, the area with significant differences covers most of the hemisphere in JJA, but, as can be seen from the analysis of the Durbin-Watson test, the summer season exhibits a large temporal autocorrelation. Hence, the statistically significant areas observed in JJA should originate from this autocorrelation. In winter, the area with significant points cover North America, another region where the Durbin-Watson test suggests serial correlation. It is clear from Fig. 2 that a possible impact, of the geomagnetic activity, if it exists, would be limited at higher latitudes, from 40° to 90°.

Because of the strong temporal autocorrelation, it is expected that at least in summer these significant differences should be false positive outcomes and they should be reduced or completely removed when applying the serial correlation correction. In fact, by applying the correction of serial dependence to the 5 hPa temperature differences the t-test results change dramatically as Fig. 3 shows. The statistically significant differences are removed everywhere in JJA. However, in DJF small areas with significant differences are still present at that level. The Durbin-Watson test somehow predicted that there will be not significant points after applying the Zweirs and von Storch algorithm in the areas where the Durbin-Watson test value was close to zero. On the other hand, the problem of multiplicity is solved here by means of the FDR procedure described in section 2. When applying such a procedure without correcting the serial dependence e some significant temperature differences still persist at 5 hPa during summertime (Fig. 4). However, the only application of FDR remove all significant differences when it is applied to other pressure levels (e.g. 10 hPa). This result is important as the FDR procedure is quite powerful in removing most of the false positive differences but, how Fig. 2 shows it is not sufficient in presence of a strong temporal correlation that can still

leave regions where the t-test rejects the null hypothesis when, in fact, it would be true. This result is particularly important and it recommends the application of both the corrections strongly.

The application of such corrections dealing both with temporal and spatial autocorrelation removes all the statistically signifi-
cant differences in the domain (Fig. 2b) and the t-test with the combined correction fails to reject the null hypothesis.

A similar result is obtained for all the other levels in the dataset, temperature differences at 1 and 100 hPa temperatures are shown in Fig. 5 and Fig. 6 without and with both the corrections. The application of the false discovery rate on those fields eliminates all the significant temperature differences, showing that also at those levels there is no detectable impact of geomag-
netic activity on the atmospheric temperature.

## 3.2   Zonally averaged temperature and 2 m temperatures.

Several studies have shown the possible impact of EEP or energetic particle precipitation on the observations using zonal mean temperatures (Tomikawa, 2017; Seppälä et al., 2013). Thus, we show how without any correction even the zonal mean temperature difference has areas that are statistically significant at 5% level (Fig. 7a). In particular, there are statistically significant areas in all the seasons, but spring, between 10 and 1 hPa. There are no statically significant area (Fig. 7b) after applying the two corrections that account for spatial and temporal autocorrelations.

It is natural to think that EEP would influence upper and mid-stratosphere temperatures through its impact on ozone. The results discussed in the previous sections suggest that the EPP influence on NH stratospheric temperatures is problematic to detect as it is much weaker than other causes of variability, among which the internal dynamical variability is paramount. As this work is motived by S09 that analyzed the 2 m temperature, Fig. 8a shows the 2m temperature difference (High Ap – Low Ap) without any correction. There are large areas where the seasonal temperature differences are statistically significant at 5% level.

The application of both the spatial and temporal autocorrelation corrections remove almost all these areas. However, some small areas of statistically significant temperature differences are still present. They are in the polar region and over Russia during the winter season and over the Scandinavia during the Spring (Fig. 8b). As it is not easy to explain these statistically significant surface temperature differences with a causal relationship with EEP, given the lack of signal aloft, there may be some other reasons that can justify this significance with other causes, among which a positive outcome obtained by chance.

## 4   Conclusions

Climate data often exhibit temporal and spatial autocorrelations which should be taken into account when testing a hypothesis, a task that is often neglected (Wilks, 2016). The effect of temporal autocorrelation was addressed with an appropriate procedure described in Zwiers and von Storch (1995) . The problem of evaluating results of multiple hypothesis tests in a spatial domain was further addressed by means of the false discovery rate procedure. In this paper, the possible impact of geomagnetic activity on the seasonal-mean stratospheric temperature in the JRA-55 reanalysis was evaluated by means a Welch's t-test under four

different cases: 1) with no correction of temporal and spatial autocorrelation, 2) with correction on temporal autocorrelation only, 3) with correction on spatial autocorrelation only, and finally 4) with both the corrections. Most of the cases examined show significant points when temporal and spatial autocorrelations are not corrected, while not showing any significant point when including just one out of the two corrections. In other words, in most cases, there is not even a need to apply both corrections to infer that there is no impact of geomagnetic activity. However, the statistically significant temperature differences at 5 hPa show that it strongly recommended the application of both the corrections for the spatial and temporal autocorrelation. In some cases, like for the JJA temperature difference at 5 hPa, there are a few significant areas remaining when applying one out of the two corrections (Figs 3 and Fig. 4), but those significant areas disappeared when both corrections were applied. Finally, the procedures to take into account these autocorrelations, the significance test typically fails to reject the null hypothesis. This result is found for all the pressure levels analyzed and for zonally averaged temperature. The only temperature field that has still statistically significant differences after applying both the corrections is the 2m temperature. There are two seasons, DJF and MAM, where small statistically significant areas are present in the polar region. In absence of a signature aloft, we therefore conclude that, based on the JRA-55 reanalyses, not enough evidence is available at present to suggest that the null hypothesis of no impact of geomagnetic activity on NH stratospheric temperatures is false. A remaining caveat concerns the definition of seasons of high or low geomagnetic activity, which is here the same as in S09 and is based on a lagged 4-month averaged Ap index, (i.e., from October to January for wintertime geomagnetic activity). Some sensitivity studies to this definition, e.g., to treat more intense shorter episodes of EEP or to treat differently the seasonal lag or accumulation of EEP, is certainly warranted for future studies. It is clear that the absence or the presence of significance does not put an end to the research of a possible relationship between EEP and stratospheric temperature, that we suppose to be weak and consequently difficult to detect.

*Data availability.* Data can be downloaded from the Meteorological Research Institute/Japan Meteorological Agency/Japan or from Research Data Archive at the National Center for Atmospheric Research, Computational and Information Systems Laboratory

*Author contributions.* NT performed the statistical tests, all the authors contributed in the interpretation of the data and wrote the paper.

*Competing interests.* The authors declare that they have no conflict of interest.

*Acknowledgements.* This work was funded by the project SOLENA - Solar effects on natural climate variability in the North Atlantic and Arctic - Research Council of Norway, Program for Space Research Project: 255276/E10. The authors acknowledge the Meteorological Research Institute/Japan Meteorological Agency/Japan for the JRA-55 reanalysis. We are extremely grateful to two reviewers for their valuable comments, corrections and suggestions.

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

**Table 1.** Years used to define the N1 set, following S09.

| Case | Hemisphere | High Ap years | Low Ap years |
|------|------------|---------------|--------------|
| N1 | NH | 1958, 1960, 1961, 1975, 1982, 1984, 1985, 1989, 1990, 1993, 1994, 1995, 2003, 2004, 2005 | 1962, 1965, 1966, 1967, 1968, 1969, 1970, 1971, 1972, 1977, 1978, 1980, 1981, 1987, 1988, 1991, 1996, 1997, 1998, 1999, 2001, 2002, 2006 |

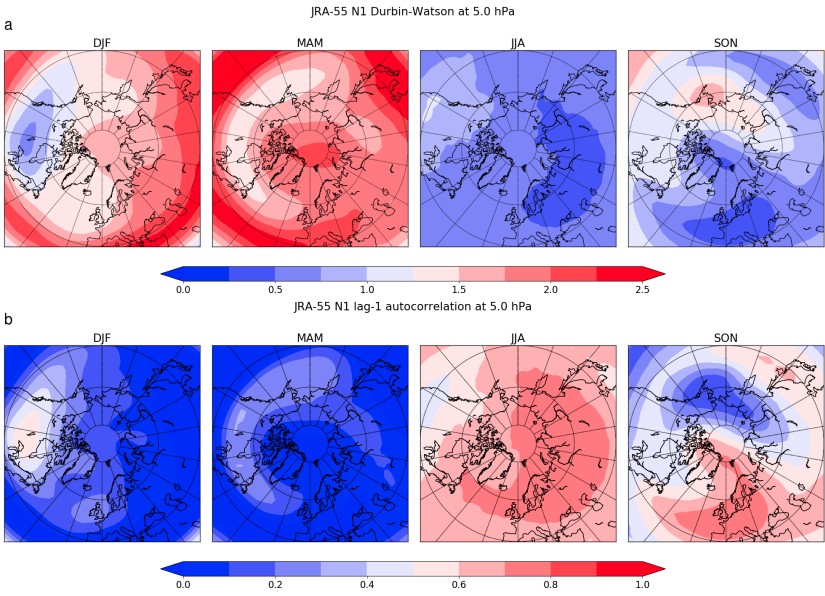

**Figure 1.** Results of Durbin-Watson test (a) and lag-1 autocorrelation (b) for JRA-55 stratospheric temperature at 5 for the period between 1958-2006.

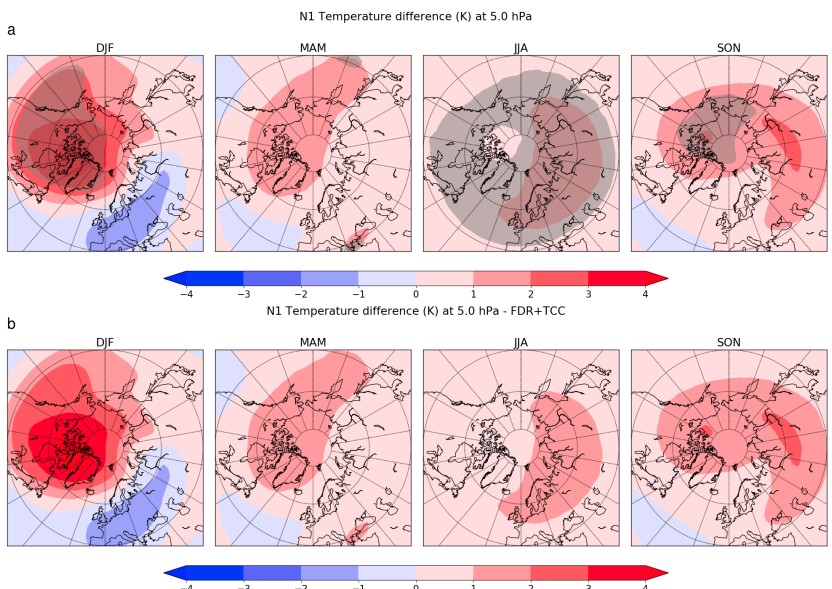

**Figure 2.** Northern hemisphere seasonal differences in stratospheric temperature ( High Ap - Low Ap ) at 5 hPa without (a) and with (b) temporal and spatial autocorrelation correction. Gray areas represent statistically significant temperature differences at the 5% confidence levels

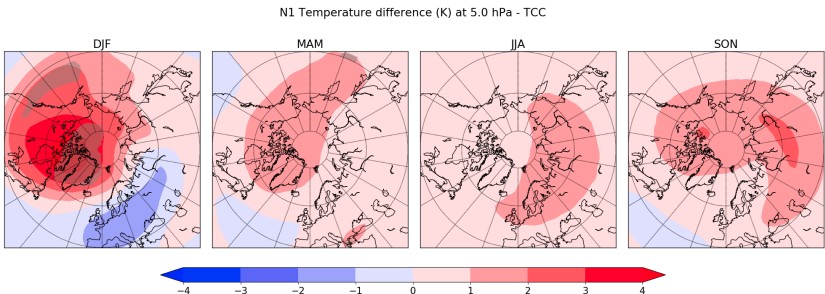

**Figure 3.** Northern hemisphere seasonal differences in stratospheric temperature ( High Ap - Low Ap ) at 5 hPa after applying the correction for serial dependence. Gray areas indicates statistically significant areas at the 5% confidence level.

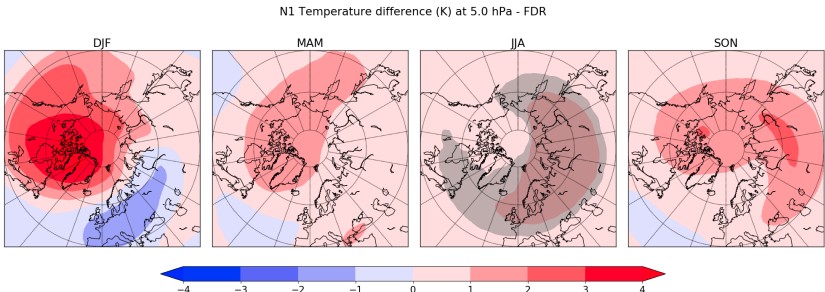

**Figure 4.** Northern hemisphere seasonal differences in stratospheric temperature ( High Ap - Low Ap ) at 5 hPa after applying the FDR correction. Gray areas indicates statistically significant areas at the 5% confidence level – before FDR correction.

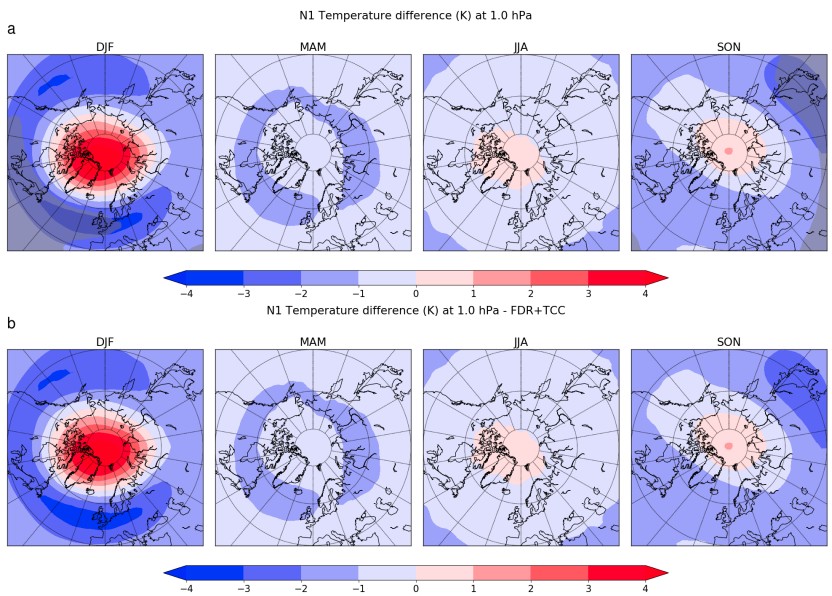

**Figure 5.** As Fig. 2 but for the 1 hPa level.

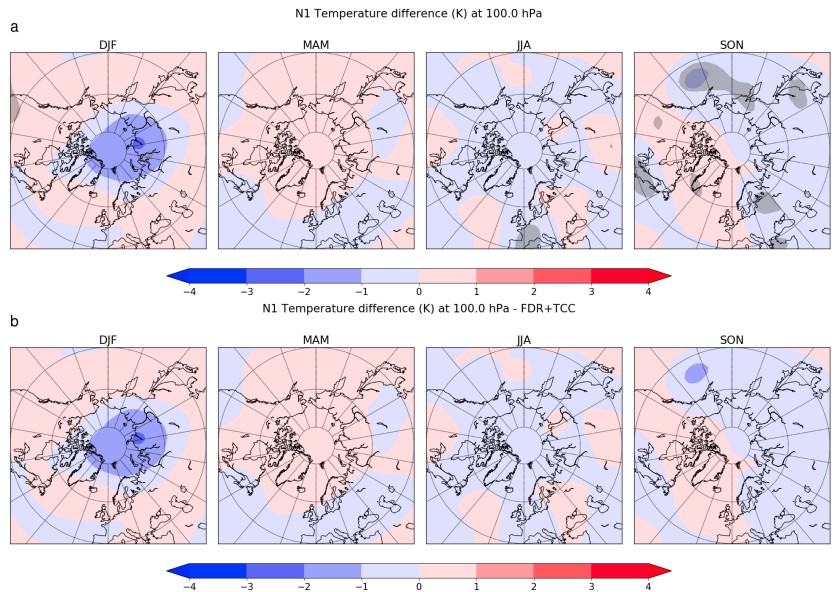

**Figure 6.** As Fig. 2 but for the 100 hPa level.

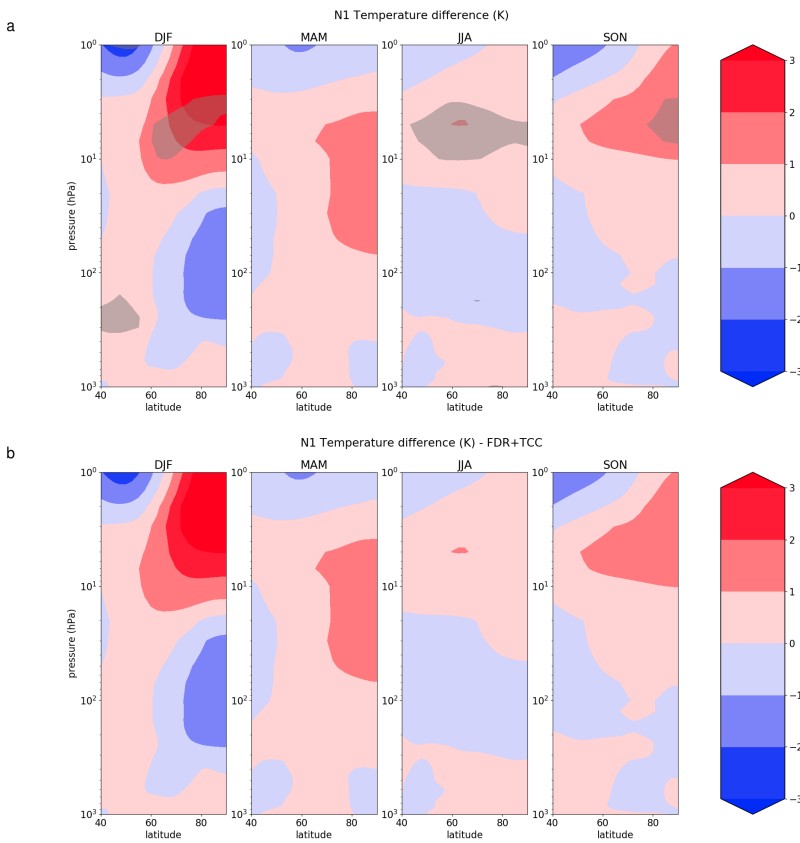

**Figure 7.** Zonal mean temperature differences (High Ap – Low Ap) without (a) and with (b) temporal and spatial autocorrelation corrections. The gray areas indicate statistically significant temperature differences at the 5% confidence level.

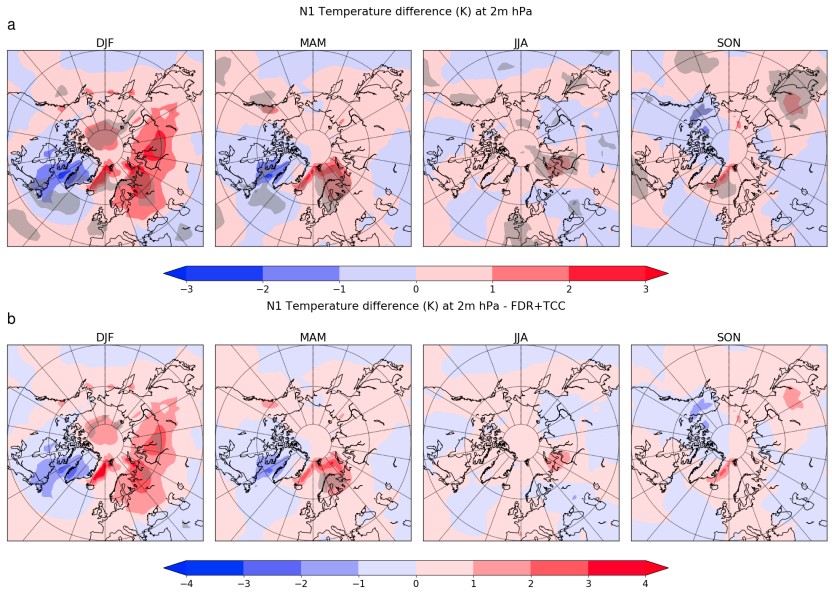

**Figure 8.** 2m temperature differences (High Ap – Low Ap) without (a) and with (b) temporal and spatial autocorrelation corrections. The gray areas indicate statistically significant temperature differences at the 5% level.