# Peer review of "A note on the statistical evidence for an influence of geomagnetic activity on JRA-55 northern hemisphere seasonal-mean stratospheric temperatures"

_Annales Geophysicae, 2019_

## Referee Comment (RC1) · Anonymous Referee #1 · 1 Jan 2020

In their manuscript, Tartaglione et al. focus on the issue of geomagnetic activity manifestations in local seasonal stratospheric temperatures. Particular attention is paid to the effects of autocorrelations and spatial relations on the formal statistical significance of temperature differences between low/high geomagnetic activity periods. The authors demonstrate that although existence of multiple regions of statistically significant temperature responses is indicated by the basic version of the t-test (i.e., version assuming mutual independence of the temperature values), these disappear when temporal autocorrelations and test multiplicity are considered.

[Figure]

By addressing the subject of statistical testing of multivariate data with notable spatiotemporal correlations, the authors tackle a vital (and arguably still rather under-investigated) area within the atmospheric research. The text is well written and structured and topically suitable for publication in Annales Geophysicae. Still, there are a few questions and formal issues that the authors should consider addressing when preparing the final version of the manuscript:

Major analysis/presentation comments:

(C1) The authors focused on point-wise analysis of temperature data and ultimately found no significant local signal related to geomagnetic activity. I wonder, however, if such signal could be more clearly seen in data averaged over larger areas, i.e. obtained for individual sectors, latitudinal bands, or over the entire extratropical area. I base this (possibly unfounded) suspicion on the presence of uniformly positive anomalies across large segments of the analysis area, notable especially for the JJA and SON seasons at the 5 hPa level (Fig. 2). By averaging the temperature series from multiple grid points, signal-to-noise ratio can perhaps be improved; conditional averages considered by the t-test may then more clearly reflect the geomagnetism influence.

(C2) The autocorrelations seem to be substantial in some of the time series. The authors address their effect through a correction reducing the number of degrees of freedom considered in the t-test. Is there, however, any identifiable source of these autocorrelations (such as a long-term trend, or imprints of solar activity variations)? If so, removal of the respective components from the time series may potentially result in higher (and statistically more significant) contrast between temperatures pertaining to low/high geomagnetic activity periods.

(C3) To quantify and visualize presence of autocorrelations in the temperature data, statistic of the Durbin-Watson (DW) test is shown in Fig. 1. Maybe presenting the lag-1 autocorrelations instead of (or in addition to) the DW statistic would better illustrate the autocorrelation structures, as they are directly involved in calculation of the corrections
applied in the paper (eq. (1)), and arguably more intuitively interpretable than the values of the DW statistic itself.

(C4) A requirement of Gaussianity is mentioned with regard to the t-test (l. 148), but, unlike other test assumptions, it is not tackled any further. I assume that this assumption is reasonably well satisfied, considering consistence of the data with AR(1) model (as discussed in the paragraph at l. 175+), but perhaps this could be mentioned explicitly?

(C5) For better comparability with topically close studies (especially Seppälä et al. (2009), by which much of the methodology in the current manuscript seems to be inspired), maybe results for lower atmospheric levels could also be shown/mentioned.

(C6) Fig. 2: The positions of grid points with statistically significant negative temperature differences (and their corresponding purple outline) seem suspicious: instead of being located within the areas pertaining to negative differences, they appear near the line separating the + and – regions

Minor/technical remarks:

l. 48: "and is thereby" to "and are thereby"

Table 1: "2001" misspelled as "20001"

l. 127: Did Seppälä et al. (2009) really use daily-step data in their analysis?

l. 141: maybe reference to Benjamini and Hochberg (1995) would be preferable here, as they are the original authors of the FDR method (as discussed later in the manuscript)

l. 219: Use of parentheses for rank-ordered p-values is introduced, but parentheses seem to be missing from the relevant formulas in the following text

l. 230: Welch's variant of the t-test is mentioned (i.e., the form assuming unequal variances of the samples compared), yet t-test employing pooled variance is presented

earlier in the text (eq. (2))

Fig. 3: The green outlines seem to be only partially drawn

l. 264: "point" to "points"

l. 300: extra comma

---

## Referee Comment (RC2) · Anonymous Referee #2 · 4 Jan 2020

General comments:

This manuscript reported a problem inherent in the attribution study of geomagnetic activity impact on the climate. The geomagnetic activity is regarded as an index for energetic electron precipitation, which attracts much attention as a third solar driver for climate. Although some previous studies suggested a strong and significant link between the geomagnetic activity and climate, its existence is still controversial. This manuscript clearly showed that the statistically significant link reported by the previous studies were likely to be erroneously detected by neglecting temporal and spatial autocorrelations of the data. This is an important suggestion and caveat for researchers dealing with this topic. The manuscript is concisely written, and the topic is suitable for AnGeo. Thus, I recommend a publication of this manuscript after addressing several points given below.

Major comments:

-Statistical significance and physical link

This study showed that the stratospheric temperature response to the geomagnetic activity was not statistically significant. Although it may be due to no physical link between them, it may be due to insufficient data length or too large internal temperature variations. The authors should mention that statistical insignificance does not deny an existence of the physical link.

-Zonal-mean temperature

Although this study is motivated by S09, the analyzed pressure levels were different (i.e., surface in S09 and stratosphere in this study). On the other hand, several previous studies examined geomagnetic activity impacts on stratosphere temperature, but only for zonal-mean temperature to my knowledge. In order to clarify whether this result can be applied to zonal-mean fields or not, I recommend showing the result for zonal-mean temperature in addition to the horizontal distribution.

-Ap index and F10.7

In this study (and S09), the Ap index was used to distinguish high and low geomagnetic activity years. Is there a potential that the correlation between Ap index and solar activity (i.e., F10.7) affects the result?

-Data length

In this study, the data between 1958-2006 was used to compare the result with S09. If the data period is extended to 2018 or 2019, does it affect the result?

Minor comments:

-p.1, l.11-21

Previous studies are not adequately cited. At least, references about energetic particle precipitation into the thermo/mesosphere and long lifetime of polar-night NOx should be added.

-p.2, l.29

"Forecast" -> "Forecasts"

-p.2, l.55

"2005" -> "2015"

-p.3, l.65

"of S09" -> "as S09"

-p.3, l.82 and p.6, l.175

"Wilks (2016)" -> "(Wilks, 2016)"

-p.4, l.91

"use" -> "use of"

-p.4, l.95

Why were 10 and 5 hPa levels chosen? While 10 hPa is representative of middle stratosphere, it seems that 1 and 100 hPa levels are appropriate as representative levels of upper and lower stratosphere, respectively.

-p.4, l.109-110

Why the AR(1) process is suitable for explaining a cumulative impact is not clear to me. Please explain it in more detail.

-p.5, l.143

"equatl" -> "equal"

-p.8

Lu et al. and Long et al. should be reversed in order.

-p.10

"20001" -> "2001"

-p. 12-14

Units in temperature should be added.

---

## Referee Comment (RC3) · Anonymous Referee #1 · 6 Jan 2020

**Referee's remark:** It seems that I based my original review on the raw manuscript file submitted by the authors on 15 Nov 2019 rather than its AnGeo-formatted version – please see below for a corrected version of my review, with line numbers consistent with the discussion paper:

In their manuscript, Tartaglione et al. focus on the issue of geomagnetic activity manifestations in local seasonal stratospheric temperatures. Particular attention is paid to the effects of autocorrelations and spatial relations on the formal statistical significance of temperature differences between low/high geomagnetic activity periods. The authors

demonstrate that although existence of multiple regions of statistically significant temperature responses is indicated by the basic version of the t-test (i.e., version assuming mutual independence of the temperature values), these disappear when temporal autocorrelations and test multiplicity are considered.

By addressing the subject of statistical testing of multivariate data with notable spatiotemporal correlations, the authors tackle a vital (and arguably still rather under-investigated) area within the atmospheric research. The text is well written and structured and topically suitable for publication in Annales Geophysicae. Still, there are a few questions and formal issues that the authors should consider addressing when preparing the final version of the manuscript:

Major analysis/presentation comments:

(C1) The authors focused on point-wise analysis of temperature data and ultimately found no significant local signal related to geomagnetic activity. I wonder, however, if such signal could be more clearly seen in data averaged over larger areas, i.e. obtained for individual sectors, latitudinal bands, or over the entire extratropical area. I base this (possibly unfounded) suspicion on the presence of uniformly positive anomalies across large segments of the analysis area, notable especially for the JJA and SON seasons at the 5 hPa level (Fig. 2). By averaging the temperature series from multiple grid points, signal-to-noise ratio can perhaps be improved; conditional averages considered by the t-test may then more clearly reflect the geomagnetism influence.

(C2) The autocorrelations seem to be substantial in some of the time series. The authors address their effect through a correction reducing the number of degrees of freedom considered in the t-test. Is there, however, any identifiable source of these autocorrelations (such as a long-term trend, or imprints of solar activity variations)? If so, removal of the respective components from the time series may potentially result in higher (and statistically more significant) contrast between temperatures pertaining to low/high geomagnetic activity periods.
(C3) To quantify and visualize presence of autocorrelations in the temperature data, statistic of the Durbin-Watson (DW) test is shown in Fig. 1. Maybe presenting the lag-1 autocorrelations instead of (or in addition to) the DW statistic would better illustrate the autocorrelation structures, as they are directly involved in calculation of the corrections applied in the paper (eq. (1)), and arguably more intuitively interpretable than the values of the DW statistic itself.

(C4) A requirement of Gaussianity is mentioned with regard to the t-test (l. 88), but, unlike other test assumptions, it is not tackled any further. I assume that this assumption is reasonably well satisfied, considering consistence of the data with AR(1) model (as discussed in the paragraph at l. 104+), but perhaps this could be mentioned explicitly?

(C5) For better comparability with topically close studies (especially Seppälä et al. (2009), by which much of the methodology in the current manuscript seems to be inspired), maybe results for lower atmospheric levels could also be shown/mentioned.

(C6) Fig. 2: The positions of grid points with statistically significant negative temperature differences (and their corresponding purple outline) seem suspicious: instead of being located within the areas pertaining to negative differences, they appear near the line separating the + and – regions

Minor/technical remarks:

l. 19: "and is thereby" to "and are thereby"

Table 1: "2001" misspelled as "20001"

l. 74-75: Did Seppälä et al. (2009) really use daily-step data in their analysis?

l. 85: maybe reference to Benjamini and Hochberg (1995) would be preferable here, as they are the original authors of the FDR method (as discussed later in the manuscript)

l. 144: Shouldn't there be J rather than i in the numerator of the fraction?

l. 146: Welch's variant of the t-test is mentioned (i.e., the form assuming unequal

variances of the samples compared), yet t-test employing pooled variance is presented earlier in the text (eq. (2))

Fig. 3: The green outlines seem to be only partially drawn

l. 163: "point" to "points"

l. 190: extra comma

---

## Author Comment (AC1) · 12 Feb 2020

Major analysis/presentation comments:

(C1) The authors focused on point-wise analysis of temperature data and ultimately found no significant local signal related to geomagnetic activity. I wonder, however, if such signal could be more clearly seen in data averaged over larger areas, i.e. obtained for individual sectors, latitudinal bands, or over the entire extratropical area. I base this (possibly unfounded) suspicion on the presence of uniformly positive anomalies across

large segments of the analysis area, notable especially for the JJA and SON seasons at the 5 hPa level (Fig. 2). By averaging the temperature series from multiple grid points,signal-to-noise ratio can perhaps be improved; conditional averages considered by the t-test may then more clearly reflect the geomagnetism influence.

Answer: As the referee 2 made a similar comment, we will add a figure with the zonal average (FIGURE 1). Averaging zonally and applying both the corrections we do not have any significant area.

(C2) The autocorrelations seem to be substantial in some of the time series. The authors address their effect through a correction reducing the number of degrees of freedom considered in the t-test. Is there, however, any identifiable source of these autocorrelations (such as a long-term trend, or imprints of solar activity variations)? If so, removal of the respective components from the time series may potentially result in higher (and statistically more significant) contrast between temperatures pertaining to low/high geomagnetic activity periods.

Answer: In general, autocorrelation is mainly due to persistence of temperature patterns year by year. For instance this is the case of the large value of temperature autocorrelation found during the summer season. However local higher autocorrelation values during other seasons can also be due also to the low frequency variance caused by large scale teleconnections (see for example Madden, 1976). The Zwiers and von Storch method is considered a standard procedure to deal with temporal autocorrelation.

(C3) To quantify and visualize presence of autocorrelations in the temperature data, statistic of the Durbin-Watson (DW) test is shown in Fig. 1. Maybe presenting the lag-1 autocorrelations instead of (or in addition to) the DW statistic would better illustrate the autocorrelation structures, as they are directly involved in calculation of the corrections applied in the paper (eq. (1)), and arguably more intuitively interpretable than the values of the DW statistic itself.

Answer: We use the Durbin-Watson because it rules out the possibility that sampling error can cause the autocorrelation. If we instead would have used lag-1 autocorrelation we would need to check whether the lag-1 autocorrelaton is statistically significant, a task that DW test does automatically. However, the plot of the lag-1 autocorrelation looks like to that obtained with DW (FIGURE 2).

(C4) A requirement of Gaussianity is mentioned with regard to the t-test (l. 88), but, unlike other test assumptions, it is not tackled any further. I assume that this assumption is reasonably well satisfied, considering consistence of the data with AR(1) model (as discussed in the paragraph at l. 104+), but perhaps this could be mentioned explicitly?

Answer: We checked the normality of the distributions over the domain and yes, in general, this condition is satisfied in most of the area. However it is widely but incorrectly believed that the t-test is valid only for normally distributed outcomes. Efron (1969) for example has shown that the t-test is still valid under weaker assumptions. We will specify this point in the revised text. point.

(C5) For better comparability with topically close studies (especially Seppälä et al.(2009), by which much of the methodology in the current manuscript seems to be inspired), maybe results for lower atmospheric levels could also be shown/mentioned.

Answer: We agree with the reviewer and we will show the 2m temperature (FIGURE 3). After the two correction steps, there is still a small area with statistically significant difference over the Scandinavia. However, this only tells us that by this feature (warm Barents Sea) the selected years are unusual. The lack of a recognizable link with stratospheric anomalies makes it difficult to establish that it is related to the geomagnetic activity.

(C6) Fig. 2: The positions of grid points with statistically significant negative temperature differences (and their corresponding purple outline) seem suspicious: instead of being located within the areas pertaining to negative differences, they appear near the line separating the + and – regions

Answer: Thank you for this observation. There was indeed a small bug in the plot subroutine that produced such features. The new figures do not have this problem as we use gray shading for significant, at the 5% level, temperature differences (FIGURE 3).

Minor/technical remarks:

l. 19: "and is thereby" to "and are thereby

"Table 1: "2001" misspelled as "20001"

l. 74-75: Did Seppälä et al. (2009) really use daily-step data in their analysis?

Answer: Yes; we guessed it when we looked at their figures, and it was confirmed by a personal communication.communication.

l. 85: maybe reference to Benjamini and Hochberg (1995) would be preferable here, as they are the original authors of the FDR method (as discussed later in the manuscript)

We agree with the reviewer.

l. 144: Shouldn't there be J rather than i in the numerator of the fraction?

J is the index representing the max value of the sorted p(i) values . We will rewrite the statement to make it clearer.

l. 146: Welch's variant of the t-test is mentioned (i.e., the form assuming un equavariances of the samples compared), yet t-test employing pooled variance is presented earlier in the text (eq. (2))

Answer: The Welch's test is always applied, even when analysing the original temperatures with n there is no correction. What is specified in equation 2 is the correction of the t-test with the Zwiers and von Storch method. In this case the pooled variance is computed with the equivalent values of n and m.

Fig. 3: The green outlines seem to be only partially drawn. 163: "point" to "points"

Answer: We solve this issue using a gray area (see Figures 3).

l. 190: extra comma

Answer: We thank the reviewer for these corrections. Most of the typos occurred when the text was converted from word to latex and they will all be corrected in the revised manuscript..

References

Bradley Efron (1969) Student's t-Test Under Symmetry Conditions. Journal of the American Statistical Association.Vol. 64, 1278-1302

Madden, R.A., 1977: Estimates of the Autocorrelations and Spectra of Seasonal Mean Temperatures over North America. Mon. Wea. Rev., 105, 9–18, https://doi.org/10.1175/1520-0493(1977)105<0009:EOTAAS>2.0.CO;2

N1 Temperature difference (K)

N1 Temperature difference (K) - FDR+TCC

**Fig. 1.** Zonally averaged temperature difference (High Ap - Low Ap) without corrections (upper panel) and with corrections (lower panel). Gray areas indicate significant differences at 0.05 level.

JRA-55 N1 Durbin-Watson at 10.0 hPa

DJF          MAM          JJA          SON

0.0    0.5    1.0    1.5    2.0    2.5

JRA-55 N1 largest autocorrelation at 10.0 hPa

DJF          MAM          JJA          SON

0.0    0.2    0.4    0.6    0.8    1.0

**Fig. 2.** 10 hPa Temperature - Durbin Watson test result (upper panel) and lag-1 autocorrelation (lower panel) for 10 hPa temperature

[Figure]

[Figure]

**Fig. 3.** 2m temperature difference (High Ap - Low Ap) after spatial and temporal autocorrelation temperature. Gray areas indicate significant differences at 0.05 level.

---

## Author Comment (AC2) · 12 Feb 2020

We want to thank the reviewers for their comments. Reply to referee2

Major comments:-Statistical significance and physical link

This study showed that the stratospheric temperature response to the geomagnetic activity was not statistically significant. Although it may be due to no physical link between them, it may be due to insufficient data length or too large internal temperature variations. The authors should mention that statistical insignificance does not deny an

existence of the physical link.

Answer: We agree with the reviewer. Our point in this paper is that often the relationship between geomagnetic activity and climate is overstated and over interpreted. However, this does not necessarily means that such a relationship does not exist at all. We will include this important remark in the conclusions.

-Zonal-mean temperature

Although this study is motivated by S09, the analyzed pressure levels were different (i.e., surface in S09 and stratosphere in this study). On the other hand, several previous studies examined geomagnetic activity impacts on stratosphere temperature, but only for zonal-mean temperature to my knowledge. In order to clarify whether this result can be applied to zonal-mean fields or not, I recommend showing the result for zonal-mean temperature in addition to the horizontal distribution.-

Answer: We have analyzed the zonal mean temperature as suggested by the reviewer, but this new analysis only confirms the results presented in the discussion version of the paper (see Figure 1)

Ap index and F10.7. In this study (and S09), the Ap index was used to distinguish high and low geomagnetic activity years. Is there a potential that the correlation between Ap index and solar activity (i.e., F10.7) affects the result?-Data length In this study, the data between 1958-2006 was used to compare the result with S09. If the data period is extended to 2018 or 2019, does it affect the result?

Answer: This comment is very interesting. The choice of the years with Ap index was taken from a previous paper (S09). However, we do not want to mix our choice with that used by S09. The relationship between Ap, F10.7 and climate is one of the topics of a separate modelling study on which we are currently working.

Minor comments: -p.1, l.11-21 Previous studies are not adequately cited. At least, references about energetic particle precipitation into the thermo/mesosphere and long

lifetime of polar-night NOx should be added.

Answer: Additional references about the role of EPP will be added.

-p.2, l.29 "Forecast" -> "Forecasts" -p.2, l.55 "2005" -> "2015" -p.3, l.65"of S09" -> "as S09" -p.3, l.82 and p.6, l.175"Wilks (2016)" -> "(Wilks, 2016)" -p.4, l.91"use" -> "use of"

-p.4, l.95 Why were 10 and 5 hPa levels chosen? While 10 hPa is representative of middle stratosphere, it seems that 1 and 100 hPa levels are appropriate as representative levels of upper and lower stratosphere, respectively.

Answer: We can show more levels, but the choice of 5 and 10 hPa was chosen primarily due to the high number of significant points present at levels 5, 7 and 10 hPa. Other levels show only a few significant points that are all removed after the application of the temporal and spatial autocorrelation. However, we can add analysis at 1 and 100 hPa, and maybe discuss the 2 m temperature difference as suggested by the referee 1, in the revised manuscript.

-p.4, l.109-110 Why the AR(1) process is suitable for explaining a cumulative impact is not clear to me. Please explain it in more detail.

Answer: We do not explicitly try to establish this relationship in the text. We discuss these two points separately.: We choose seasonal temperature as we think that EEP, differently from proton events, has a cumulative impact; then we assume that seasonal temperature can be treated as an AR(1) process in order to apply the method of Zwiers and von Storch. The use of an AR(1) process for describing or modelling seasonal temperature is quite common in climate research (e.g. Wakaura and Okata, 2007).

p.5, l.143"equatl" -> "equal"- p.8 Lu et al. and Long et al. should be reversed in order.-p.10"20001" -> "2001"- p. 12-14Units in temperature should be added.

Answer: We thank the reviewer for all the corrections. Most of the typos occurred when the text was converted from word to latex and they will all be corrected in the revised manuscript..

**ANGEOD**

References:

Wakaura, M. and Ogata, Y. (2007), A time series analysis on the seasonality of air temperature anomalies. Met. Apps, 14: 425-434. doi:10.1002/met.41
[Figure]

N1 Temperature difference (K)

N1 Temperature difference (K) - FDR+TCC

**Fig. 1.** Zonally averaged temperature difference (High Ap - Low Ap) without corrections (upper panel) and with corrections (lower panel). Gray areas indicate significant differences at 0.05 level.

---

## Author Comment (AC3) · 20 Feb 2020

We are not sure we have fully answered the following point: (C2) removal of the respective components from the time series may potentially result in higher (and statistically more significant) contrast between temperatures pertaining to low/high geomagnetic activity periods.

Answer: It is not clear to us what the reviewer means with "removal of the respective components from the time series". The used procedure does not remove any data

neither performed we a sub-sampling of data. In such a case, we would throw away most of our data and important information. Instead, as the serial correlation leads to overestimates of statistical significance, the effective number of degrees of freedom can be much smaller than the sample size would indicate. What the procedure does to deal with the serial correlation is to account for this loss of degrees of freedom by calculating the "effective" sample size, that is the number of independent data.

---

## Author Response (AR1)

Reviewer 1

Major analysis/presentation comments:

(C1) The authors focused on point-wise analysis of temperature data and ultimately found no significant local signal related to geomagnetic activity. I wonder, however, if such signal could be more clearly seen in data averaged over larger areas, i.e. obtained for individual sectors, latitudinal bands, or over the entire extratropical area. I base this(possibly unfounded) suspicion on the presence of uniformly positive anomalies across large segments of the analysis area, notable especially for the JJA and SON seasons at the 5 hPa level (Fig. 2). By averaging the temperature series from multiple grid points,signal-to-noise ratio can perhaps be improved; conditional averages considered by the  t-test may then more clearly reflect the geomagnetism influence.

**In order to answer to both the reviewers we added the section 3.2 to discuss the zonally averaged temperature differences and the new Figure 7**

(C2) The autocorrelations seem to be substantial in some of the time series. The authors address their effect through a correction reducing the number of degrees of freedom considered in the t-test. Is there, however, any identifiable source of these autocorrelations (such as a long-term trend, or imprints of solar activity variations)? If so, removal of the respective components from the time series may potentially result in higher (and statistically more significant) contrast between temperatures pertaining to low/high geomagnetic activity periods.

**There may be many causes for autocorrelation. We added a short discussion, see lines 110-115.**

(C3) To quantify and visualize presence of autocorrelations in the temperature data,statistic of the Durbin-Watson (DW) test is shown in Fig. 1. Maybe presenting the lag-1autocorrelations instead of (or in addition to) the DW statistic would better illustrate the autocorrelation structures, as they are directly involved in calculation of the corrections applied in the paper (eq. (1)), and arguably more intuitively interpretable than the values of the DW statistic itself.

**We added the lag-1 autocorrelation together the DW test result (see new Fig. 1)**

(C4) A requirement of Gaussianity is mentioned with regard to the t-test (l. 88), but, un-like other test assumptions, it is not tackled any further. I assume that this assumptionis reasonably well satisfied, considering consistence of the data with AR(1) model (asdiscussed in the paragraph at l. 104+), but perhaps this could be mentioned explicitly?

**We discussed this issue at lines 91-93 of the new manuscript.**

*It is however well known that the t-test, which assumes a statistical model where observations are statistically independent and it is widely, but incorrectly, believed that the t-test is valid only for normally distributed outcomes. Several authors (Efron, 1969; De Winter, 2013; Poncet et al., 2016) have shown that the t-test is suitable under symmetric, not necessarily normal, and asymmetric distributions.*

(C5) For better comparability with topically close studies (especially Seppälä et al.(2009), by which much of the methodology in the current manuscript seems to beinspired), maybe results for lower atmospheric levels could also be shown/mentioned.

**We discussed this point in the new section 3.2 and showing the results  in Fig. 8.**

(C6) Fig. 2: The positions of grid points with statistically significant negative temperature differences (and their corresponding purple outline) seem suspicious: instead of being located within the areas pertaining to negative differences, they appear near thel ine separating the + and – regions

**CORRECTED**

Minor/technical remarks:l. 19: **CORRECTED**
"and is thereby" to "and are thereby" **CORRECTED**
Table 1: "2001" misspelled as "20001" **CORRECTED**
l. 74-75: Did Seppälä et al. (2009) really use daily-step data in their analysis?

l. 85: maybe reference to Benjamini and Hochberg (1995) would be preferable here, as they are the original authors of the FDR method (as discussed later in the manuscript) **CORRECTED**

l. 144: Shouldn't there be J rather than i in the numerator of the fraction? **CORRECTED It is not necessary to define J.**

l. 146: Welch's variant of the t-test is mentioned (i.e., the form assuming unequal variances of the samples compared), yet t-test employing pooled variance is presented earlier in the text (eq. (2))

**We discussed this issue in the interactive discussion. The Welch's test is always applied even when there is no correction, we say that at line 72.**

Fig. 3: The green outlines seem to be only partially drawn **CORRECTED. Now there is a gray area where differences are significant.**

l. 163: "point" to "points"l. 190: extra comma **CORRECTED**
* * *
Reviewer 2

Major comments:
-Statistical significance and physical link
This study showed that the stratospheric temperature response to the geomagnetica ctivity was not statistically significant. Although it may be due to no physical link between them, it may be due to insufficient data length or too large internal temperature variations. The authors should mention that statistical insignificance does not deny an existence of the physical link.

**We stress this point in the new manuscript at pag. 7 – lines 197-199**
*It is natural to think that EEP would influence upper and mid-stratosphere temperatures through its impact on ozone. The results discussed in the previous sections suggest that the EPP influence on NH stratospheric temperatures is problematic to detect as it is much weaker than other causes of variability, among which the internal dynamical variability is paramount.*

**And at the end of conclusions, pag. 8 lines 231-232**
I*t is clear that the absence or the presence of significance does not put an end to the research of a possible relationship between EEP and stratospheric temperature, that we suppose to be weak and consequently difficult to detect.*

-Zonal-mean temperatureAlthough this study is motivated by S09, the analyzed pressure levels were different(i.e., surface in S09 and stratosphere in this study). On the other hand, several previous studies examined geomagnetic activity impacts on stratosphere temperature, but only for zonal-mean temperature to my knowledge. In order to clarify whether this result canbe applied to zonal-mean fields or not, I recommend showing the result for zonal-mean temperature in addition to the horizontal distribution.

**This suggestion was implemented adding the section 3.2 at pag. 7 and new figure 7.**

-Ap index and F10.7
In this study (and S09), the Ap index was used to distinguish high and low geomagneticactivity years. Is there a potential that the correlation between Ap index and solar activity (i.e., F10.7) affects the result?-

Data length. In this study, the data between 1958-2006 was used to compare the result with S09. If the data period is extended to 2018 or 2019, does it affect the result?

**As we already said in the interactive discussion we have a submitted paper based on sensitivity experiments where we face these questions. We can share here our conclusions that Solar activity alone has a significant impact on the ozone chemistry and on mesospheric and stratospheric temperature, whereas GA hasn't. However, there is a mutual interaction between SSI and GA and this happens when they are both high and act together.**
**In our opinion, observations do not allow to evaluate such complex interactions.**

Minor comments:
-p.1, l.11-21Previous studies are not adequately cited. At least, references about energetic particle precipitation into the thermo/mesosphere and long lifetime of polar-night NOx should be added.
**ADDED  (see lines 13-22)**
-p.2, l.29"Forecast" -> "Forecasts"    **CORRECTED**
-p.2, l.55"2005" -> "2015"  **CORRECTED**
-p.3, l.65"of S09" -> "as S09"-p.3,  **CORRECTED**
l.82 and p.6, l.175"Wilks (2016)" -> "(Wilks, 2016)"  **CORRECTED**
-p.4, l.91"use" -> "use of"  **CORRECTED**

-p.4, l.95 Why were 10 and 5 hPa levels chosen? While 10 hPa is representative of middlestratosphere, it seems that 1 and 100 hPa levels are appropriate as representativelevels of upper and lower stratosphere, respectively.

**This comment was implemented removing the 10 hPa and adding 1 and 100 hPa. We kept the 5 hPa level instead of the 10 hPa level because it teaches us that the application of the only FDR procedure sometimes is not enough and both the corrections have to be applied.**

-p.4, l.109-110 Why the AR(1) process is suitable for explaining a cumulative impact is not clear to me. Please explain it in more detail.

**We discussed this point in the interactive discussion.**

-p.5, l.143"equatl" -> "equal"   **CORRECTED**

-p.8Lu et al. and Long et al. should be reversed in order. **CORRECTED**
-p.10"20001" -> "2001" **CORRECTED**
-p. 12-14Units in temperature should be added. **ADDED**

[revised manuscript text omitted]